Label dependency modeling in Multi-Label Naïve Bayes through input space expansion

Chitra PKA 1
Balasubramanian Saravana Balaji 2
Khattab Omar 3
Al-Kadri Mhd Omar omar.alkadri@udst.edu.qa 4
1 Department of Information Technology, Rathinam Group of Institutions , Coimbatore , Tamil Nadu , India
2 Department of Computing, De Montfort University Kazakhstan , Almaty , Kazakhstan
3 Department of Computer Science and Engineering, Kuwait College of Science and Technology , Kuwait
4 College of Computing and Information Technology, University of Doha for Science and Technology , Doha , Qatar
Pires Ivan Miguel
Electronic publication date: 2024 Dec 10
Publication date: 2024
Volume: 10
Electronic Location ID: e2093
Received 2024 Feb 13; Accepted 2024 May 8
Copyright: ©2024 Chitra et al.
Copyright year: 2024
Copyright holder: Chitra et al.
License: This is an open access article distributed under the terms of the Creative Commons Attribution License, which permits unrestricted use, distribution, reproduction and adaptation in any medium and for any purpose provided that it is properly attributed. For attribution, the original author(s), title, publication source (PeerJ Computer Science) and either DOI or URL of the article must be cited.
License URL: https://creativecommons.org/licenses/by/4.0/

Keywords: Multi-label Naïve Bayesian classification, Label dependency, Input space expansion, Heterogeneous feature space, Mixed joint density distribution

Funding: The authors received no funding for this work. The Open Access APC is provided by Qatar National Library. The funders had no role in study design, data collection and analysis, decision to publish, or preparation of the manuscript.

==============================
In the realm of multi-label learning, instances are often characterized by a plurality of labels, diverging from the single-label paradigm prevalent in conventional datasets. Multi-label techniques often employ a similar feature space to build classification models for every label. Nevertheless, labels typically convey distinct semantic information and should possess their own unique attributes. Several approaches have been suggested to identify label-specific characteristics for creating distinct categorization models. Our proposed methodology seeks to encapsulate and systematically represent label correlations within the learning framework. The innovation of improved multi-label Naïve Bayes (iMLNB) lies in its strategic expansion of the input space, which assimilates meta information derived from the label space, thereby engendering a composite input domain that encompasses both continuous and categorical variables. To accommodate the heterogeneity of the expanded input space, we refine the likelihood parameters of iMLNB using a joint density function, which is adept at handling the amalgamation of data types. We subject our enhanced iMLNB model to a rigorous empirical evaluation, utilizing six benchmark datasets. The performance of our approach is gauged against the traditional multi-label Naïve Bayes (MLNB) algorithm and is quantified through a suite of evaluation metrics. The empirical results not only affirm the competitive edge of our proposed method over the conventional MLNB but also demonstrate its superiority across the aforementioned metrics. This underscores the efficacy of modeling label dependencies in multi-label learning environments and positions our approach as a significant contribution to the field.

Introduction

The advent of multi-label classification has brought forth a paradigm shift in machine learning, challenging the traditional confines of single-label datasets (Han et al., 2023). In a multi-label context, instances are inherently complex, often associated with multiple labels simultaneously, which reflects the multifaceted nature of real-world data. This complexity necessitates algorithms capable of discerning the subtle interdependencies among labels to make accurate predictions. Naïve Bayes, a stalwart in the pantheon of classification algorithms due to its simplicity and efficacy, traditionally operates under the assumption of label independence—an assumption which is starkly at odds with the multi-label environment. Our research introduces an innovative approach to multi-label Naïve Bayes (iMLNB), which not only acknowledges but also capitalizes on the dependencies between labels. By expanding the input space to include meta information from the label space, we construct a more informative and nuanced feature set. This paper delineates the development of this enhanced iMLNB model, its theoretical underpinnings, and the empirical validation of its performance across diverse datasets. In doing so, we aim to set a new benchmark for multi-label classification and open avenues for further exploration in this dynamic field.

Traditional single-label classification (SLC) algorithm associates every instance in data f exactly to single output (ℓ). SLC is defined as in Eq. (1). (1) SLCh:fi↦ℓ

where ℓ=1. By definition (Han et al., 2023), multi-label classification (MLC) is a generalization of supervised single-label classification undertaking at which just about every data example could possibly be related to a pair of class labels and each label contains binary outcomes. Let F be the set of Features, F = {f1, f2, …, fk} and L be the set of labels, L = {ℓ1, ℓ2, ℓ3….., ℓm}, the multi-label dataset D is D = {F, L}, the ℓi = 1, if the label is and relevant with an instance and ℓi =0, otherwise. The objective of multi-label classification is to map a sample instance χ ∈{D} to a label set ℓ € {£} defined as in Eq. (2) (2) MLCh:fi↦∏i=1mℓ

where ℓ≥ 2. Each label contains binary outcomes. Due to an increase in application areas wherein samples demand more than single-label, multi-label is an emerging field of machine learning. Various types of classification tasks include: (i) binary classification, (ii) multi class classification and (iii) multi-label classification. Multi-label classification is used in the field of bioinformatics document and music categorization (Huang et al., 2023; Trohidis et al., 2008), semantic image and video annotation (Zhang, 2024; Feng & Xu, 2010), drug side effect prediction (Zhang et al., 2016), species distribution modelling (Jones, Miller & White, 2011), etc. Naïve Bayes is a supervised learning method where a simple probabilistic classifier based on Bayes’ theorem is used to predict unseen data and features are assumed to be independent.

Naïve Bayes had been extended to multi-label Naïve Bayes (MLNB) under the assumption of label independence. MLNB uses conditional probability to estimate likelihood of the class label conditioned on predictive attribute space. P probability of a class for given attributes is the factorization of joint probability, as in Eq. (3). (3) HNB:X→∏1≤j≤ndPℓ1b,Pℓ2b,…Pℓ|L|b

Base value b, gives the presence or absence of label for instance in concern. ℓ11, defines presence of label ℓ1 in test instance t, and ℓ10 defines absence of same. This state-of-the-art approach ignores label dependency as the decomposition of multi-label to single-label was built on binary relevance (BR) approach. This decomposition mechanism ignores correlation among labels which may penalize performance of the methodology. BR is the baseline for multi-label classification; however, its performance tends to be poor with strong label dependencies. For example, an image elucidated with forest can also be labelled as trees; an article about a movie can be labelled as entertainment and sports can be labelled as entrainment (Fan et al., 2024). In such cases, label dependencies need to be exploited to enhance performance of multi-label classification models. Relevance of one label may depend on an other in a multi-label dataset, where labels shouldn’t be treated separately as BR does. Hence, ignorance of label correlation may lead to loss of label information. With label independence assumption, predicted label set may contain few or many unexpected labels that may never occur in the training phase.

This paper extends Naïve Bayes for multi-label classification by introducing an adaptive improved multi-label Naïve Bayes (iMLNB) that explores label interactions through input space expansion and meta information from a target space used in the expanded input space. Proposed method expands predictor set with all but one response variables which enables models to explore label correlation during learning phase; the model will then use not only the predictor information but also label information for the prediction of unseen data. This expansion causes exponential growth in predictor space with labels from label space. The growth may lead to over fitting and to solve this, meta data with richer information only are considered in input space. This expansion comprises a mixture of heterogeneous predictors (both continuous and discrete features) in feature space, which causes multi-label Naïve Bayes to be adopted to deal with both discrete and continuous predictors directly.

Principal contributions of the paper

The proposed work addresses label correlation ignorance in multi-label Naïve Bayes and is addressed by incorporating labels learnt in (i-1)th iteration into ith iteration as meta data. Later, over fitting caused by the inclusion has been solved by choosing richer features. The following are principal contributions in this research work:

• Input Space Expansion: The research addresses the common issue of label dependency ignorance in multi-label Naïve Bayes (MLNB) by expanding the input space. This is achieved by incorporating meta-information from the (i-1)th iteration into the ith iteration, utilizing all but one label information to capture label correlations.

• Overfitting Mitigation: To counteract overfitting resulting from the expanded input space, the approach includes a feature selection process that focuses on highly correlated labels, thereby enhancing the model’s generalizability.

• Hybrid Likelihood Estimation: The likelihood in the proposed iMLNB method is adapted using a combination of the cumulative distribution functions of Gaussian and Bernoulli distributions. This allows for effective handling of both continuous and discrete predictors within the heterogeneous input space.

• Empirical Validation: The proposed iMLNB method has been empirically tested across three application domains—biology, image, and text—and has shown improved performance over conventional MLNB and other multi-label algorithm adaptations.

• Benchmark Dataset Performance: The performance of the proposed iMLNB method has been validated using six multi-label benchmark datasets with varying input and label sizes, confirming its superiority over existing methods.

The paper is organized as follows: ‘Introduction’ introduces the topic and outlines the research problem. Section ‘ Literature Survey’ reviews related work in the field. Section ‘Materials and Methods’ provides background on multi-label learning, the evaluation metrics, the datasets used, and the proposed method’s formulation. Section ‘Experimental Setup’ details the experimental setup and procedures. Section ‘Results and Discussion’ presents and discusses the results. Finally, ‘Conclusion’ concludes the paper and suggests directions for future research.

Literature Survey

The exploration of multi-label methods has yielded significant insights into the transformation of data within a multi-label environment. Du et al. (2024) underscored the binary relevance (BR) method as a fundamental baseline for this transformation. BR simplifies the complexity of multi-label problems by decomposing them into individual binary classification tasks, one for each label within the label space L. This decomposition allows for each instance in the training set to be distinctly labelled as relevant or irrelevant, contingent upon its association with the respective multi-label example.

Building upon this foundation, an innovative modification to BR was introduced by Radovanović et al. (2023), termed the classifier chain (CC). This method employs a sequence of classifiers, each predicting the binary association of labels while enhancing the feature space with the labels previously predicted in the chain. Despite its efficacy, the CC method introduces an additional layer of complexity due to the necessity of maintaining the order of the chains. This complexity was adeptly managed through the implementation of ensemble methods, which aggregate the predictions of labels, allowing each label to accrue votes. A threshold is then applied to discern the relevant labels, streamlining the decision-making process.

The utility of BR, despite its limitations, was further elucidated by Luaces et al. (2012). The authors demonstrated BR’s effectiveness as a baseline method for multi-label learning, particularly for its computational efficiency and its capability to produce optimal methods when targeting macro-averaged measure-based loss functions. Their assertions were substantiated through the use of benchmark datasets as well as synthetic datasets, which also revealed BR’s competitive edge against more intricate, conventional approaches, especially in scenarios characterized by high-dimensional and highly correlated label spaces.

Addressing the often-overlooked label dependencies in BR, Alvares-Cherman, Metz & Monard (2012) extended the BR approach to effectively predict label combinations by enriching the feature space with comprehensive label information from the label space, excluding the current label under consideration. This extension facilitated a more nuanced understanding of label interdependencies, which was validated through binary learning methods.

In the domain of multi-target regression, a novel methodology was introduced that expanded the input space to accommodate multi-target datasets (Guo & Li, 2014). Their approach involved the stacking of multiple single targets and ensembles of regressor chains, adopting multi-label strategies to address the challenges posed by the inclusion of out-of-sample estimates of target variables during the training phase.

Furthermore, the ML-C4.5 algorithm represents a significant modification of the C4.5 algorithm (Moral-García et al., 2022). This adaptation recalibrates the entropy calculations for each label, allowing the algorithm to identify attributes with the highest information gain and construct a decision tree structure with bifurcated child nodes tagged with values of 1 and 0, respectively.

The landscape of multi-label learning has been significantly shaped by a series of innovative approaches that aim to accommodate the complexity of multiple labels associated with single instances. A back propagation multi-label learning (BPMLL) algorithm revolutionizes the error function to account for multiple labels (Joe & Kim, 2024). This approach revises the conventional learning algorithm to minimize error by prioritizing labels associated with a given instance and reducing the error functions, which are defined as the squared difference between the output signal values and the target values for each training case.

Furthering the evolution of multi-label learning, the conventional k-nearest neighbors (kNN) algorithm was modified to create multi-label kNN (MLkNN) (Tian et al., 2024). This adaptation leverages the presence of multiple labels by first identifying the k-closest neighbors from the training set for each test instance and then calculating a membership counting vector. The Euclidean distance metric serves as the measure between instances. The labels for unseen instances are then determined using the maximum a posteriori (MAP) principle, based on the membership counting vector.

In the realm of Naïve Bayes, a multi-label Naïve Bayes (MLNB), was suggested to extend the single-label Naïve Bayes to handle multiple labels (Wei et al., 2011). This method utilizes binary relevance transformation as a preliminary step, transforming multiple labels into a single-label format that can be processed by the traditional Naïve Bayes algorithm. The final output is a synthesis of all single-label transformations, with the Gaussian probability distribution employed for likelihood parameter estimation, particularly as their research focused on datasets with continuous attributes.

A Naïve Bayesian (NB) algorithm for multi-label classification was introduced to incorporates a two-step feature selection process to uphold the assumption of conditional independence central to Naïve Bayes classification theory (Zhang & Zhou, 2013). The authors also developed a prototype system that categorizes the results of text search queries into multiple categories.

A double weighted Naïve Bayes utilizes a niching cultural algorithm (DWNB-NCA) to automatically determine the optimal weight configuration for enhancing classification accuracy (Yan, Wu & Sheng, 2016). In this system, a population is composed of individuals that represent the weight for attributes and labels in the multi-label dataset. The niche cultural algorithm (NCA) is then employed to train selected examples from the training data to determine the most effective weight combinations.

The MLNB-LD, a multi-label Naïve Bayes algorithm acknowledges the dependencies among labels during the classification process (Lee, 2018). This method introduces a new posterior probability estimation method that relaxes the strong independence assumption typically associated with Naïve Bayes. The geometric mean is utilized to aggregate the estimations, providing a more holistic view of the label dependencies.

The field of multi-label document classification has witnessed a series of methodological advancements aimed at enhancing the precision and efficiency of multi-label classifiers. A nuanced approach to the classification of multi-label documents was pioneered by introducing fine-grained weights within the framework of multinomial Naïve Bayes (Krishnamoorthy, Sathiyanarayanan & Proença, 2024). Their methodology hinges on two pivotal techniques: the value weighting approach, which assigns weights to feature values, and the co-training technique, which capitalizes on the dependencies among class values. The empirical results from their experiments demonstrated the superiority of their method over other contemporary approaches. Table 1 presents the comparisons of different methods to capture label correlations.

Table 1 Comparison of methods to capture label correlations.

Chochlakis et al. (2023)	• Investigated masked emotion (MEmo) which was based on masked language modeling and demultiplexer (DEMUX) which leverage global relation and multiple sequences of inputs.
•  Assessed numerous configurations of global and local regularization losses that regulate the emotional responses of pairs of emotions.
•  Classification loss is addressed through global (static –the constrains based on prior relationships among emotions) and local (dynamic -all the global labels of all samples are considered) terms.	
Zhang (2024)	• Label correlations were captured by maximizing the margin between irrelevant and relevant labels.
•  Created a MLRPA –Multi-label passive aggressive classification algorithm that model an error set by predicting couple of labels.	
Fan et al. (2024)	• A model named Learning Correlation Information for Multi-label Feature Selection (LCIFS)
•  Adopted a three-step process to select features that has significant correlation with label.
•  A low dimensional representation of feature space was created by mapping structural correlations of label space through adaptive spectral graph in low dimensional space.	
Du et al. (2024)	• Developed a model named Semi-supervised imbalanced multi-label classification with label propagation (SMCLP)
•  Learns a label regularization matrix for an labeled instance and then incorporated the relationship into loss function.
•  An weighted undirected graph is used to find similarity among all instances of labeled and unlabeled data.	

In a novel stride, a directed acyclic graph (DAG), ordered multi-label classification approach was introduced to leverage the canonical correlated autoencoder as a classifier and introduces the DAGLabel approach to address the constraints imposed by DAG ordering (Feng, Zhao & Fu, 2020). The DAGLabel method was shown to improve classification accuracy by meeting the hierarchical restrictions often required in complex datasets, such as human gene information annotated with gene ontology.

A granular computing perspective to analyze the distribution among labels and introduce a unique multi-label attribute selection algorithm based on label distribution and feature complementarity was introduced by Qian et al. (2020). Their method’s effectiveness was validated through extensive experiments on ten open-source multi-label datasets, outperforming other modern techniques across six widely used measures.

Non-equilibrium based multi-label learning method not only considers the classification margin but, also emphasizes the identification of missing labels (Cheng et al., 2020). The authors introduced a category margin concept, expanded the label space, and employed information entropy to measure the inter-label connections, culminating in the creation of a label confidence matrix. The experimental results underscored the method’s advantages over other multi-label learning algorithms.

The reassignment of misclassified features with new weights through weighted information gain was focused to enhance classification accuracy (Kaur, Balakrishnan & Wong, 2022). The method concentrated on the weights of frequency bins to represent target word features and was trained on a corpus extracted from Facebook pages. The weighted information gain approach was then applied in conjunction with multinomial Naïve Bayes to capture label dependencies, demonstrating the growing importance of social media as a rich data source for sentiment analysis.

The burgeoning volume of social media usage and the consequent increase in data generation was assessed by Rani et al. (2023). The authors highlighted the importance of pre-processing techniques such as stemming and stop word elimination in sentiment analysis, with their methodology showing that models employing these techniques, particularly in conjunction with multi-class Naïve Bayes, significantly outperformed others.

Lastly, the performance comparison between artificial neural networks (ANN), multinomial Naïve Bayes, and support vector machine (SVM) approaches in data classification was addressed by Rani et al. (2023). The authors selected ANN for its versatility across various fields and tasks, including data generation, classification, and regression. Their research utilized diverse datasets, ranging from single-label to multi-label, and employed k-fold validation to ensure the robustness and reproducibility of the model’s results, with performance metrics such as the F1-score serving as benchmarks for evaluation. The ability to identify and interpret emotions conveyed through written text has become crucial in various domains. Chochlakis et al. (2023) examined methods to leverage label correlations in multi-label emotion recognition models in order to enhance emotion identification. They devised two modelling methodologies to capture word connections related to emotions. These methodologies involve incorporating emotions into the input and using pairwise constraints of emotion representations as regularization terms. De Lima et al. (2022) have used bidirectional encoder representations form transformers for the classification of tax goods that were official classification systems for trading the products in Brazil.

The proposed research endeavours to advance the capabilities of the MLNB algorithm by introducing an improved variant, herein referred to as the iMLNB. The cornerstone of this enhancement lies in the explicit incorporation of label dependencies, achieved by expanding the feature space to include —L—-1 labels. This expansion results in a feature space characterized by heterogeneity, encompassing a mix of both continuous and discrete data types. To adeptly manage this diversified input space, we have tailored the parameters of the Naïve Bayes algorithm within the iMLNB framework. These modifications are meticulously designed to accommodate the complexities introduced by the additional label information, ensuring that the algorithm can still perform its probabilistic estimations with precision. The methodologies that underpin this research, as well as the anticipated efficacy of the proposed iMLNB method, are set to be expounded in the forthcoming sections of our work. We will delve into the specific techniques employed to modify the Naïve Bayes parameters and discuss the theoretical underpinnings that justify these changes. Furthermore, we will outline the experimental setup designed to evaluate the performance of the iMLNB method, providing a comparative analysis against standard MLNB approaches.

Materials and Methods

Background

Multi-label learning and evaluation metrics

In the realm of multi-label learning, the complexity of the problem is significantly heightened when compared to traditional single-label learning scenarios. The foundational concept of multi-label learning involves instances that are inherently associated with multiple labels simultaneously, rather than a single label. This section will elucidate the formal definition of the multi-label learning problem and delineate the evaluation metrics employed to assess the performance of multi-label classifiers. Let dataset D = {x1, x2, …, xk}, where each xi (for i = 1,2, …, k) is characterized by a set of predictive attributes, labels L = {l 1, l 2, …, l m} where each l, j (for j = 1, 2, …, m), represents a distinct label from the response space. Multi-label classifier 𝔥 is defined as 𝔥: f → 2  l. Each instance in example is related to more than one label from label space. Evaluation metrics used to evaluate multi-label learning are more complicated than traditional single-label learning. Commonly used evaluation metrics are described below:

Hamming_Loss ↓.

It requires the error of prediction, analysis of missing error into consideration and testimonials typical example-label set misclassification. Lower h_loss value shows higher classifier performance. The hamming_Loss is specified in Eq. (4). (4) h_lossxi,yi=1q∑i=1q1łhxi△li

△ is XOR operation i.e. symmetric difference between two sets and l defines count of labels.

Or

Let tpi, fpi tni, and fni are true positive, false positive and true negative false negative of ith sample and h_loss can be defined as in Eq. (5). (5) h_lossxi,yi=1q∑i=1qfpi+fnitpi+fpi+tni+fni

One_error ↓.

It quantifies the lack of highly ranked labels against relevant label in the training set; with values ranging from 0 to 1 and is similar to classification error in conventional methods. Smaller one_error value describes better performance and is described in Eq. (6). (6) One_errorħ=1N∑i=1NargmaxλɛLxiλ

Ranking loss ↓.

For the provided example, it calculates the mean of label pairs that are arranged in reverse; and is presented in Eq. (7). (7) RankingLoss=1N∑i=1Nla,lbxilxil¯

Where rank of la is greater than lb and la,lb∈xilX xil¯ and the algorithm performs better when the ranking loss is less.

Average precision ↑.

It figures out the average ratio of significant labels in each label pair before averaging all relevant labels. It is defined in Eq. (8), (8) AveragePrecision=1N∑i=1N1xil ∑λ∈xilλ′∈xilriλ

where riλ′≤riλ.The learning algorithm performs better with a higher average precision value, and when average precision equals 1, the method performs flawlessly.

Subset accuracy ↑.

The Jaccard similarity coefficient among label sets (x i) and (y i) provides as its definition. This assessment is an average across all instances. The description is given in Eq. (9). (9) Subset_accuracy=1N∑i=1Nħxi∩yiħxi∪yi

Binary relevance approach basics

The process of learning from multi-label data involves breaking the multi-label problem down into a number of single-label problems, then learning using a number of base classifiers. Binary relevance (BR) is the most commonly used baseline transformation method because of its simplicity and effectiveness. It has many advantages: (i) simple and independent and (ii) low computational complexity. The binary relevance method uses one–versus–rest strategy and initially transforms the source dataset D into L datasets Dli for each li ∈L labels and is defined in Eq. (10). (10) Dj′=xi,li,ifliLli,otherwise

Each transformed dataset Dli consists of all examples from source datasets with relevant (positive) or irrelevant (negative) labels associated with single-label li, while labels other than li are excluded, any supervised binary learning method can be used as the base algorithm. The BR method learns —L— individual {HBR1 (D1′, HBR2 D2′, …, HBR—L—(D|L|′} binary classifiers (C li) over the transformed dataset Dli for each label in label space. Individual binary supervised classifier works as in Eq. (11). (11) HBRjDj′:C liX→li|¬li∀li∈L;li∈0,1

The L classifiers are completely independent, implying that the prediction of one classifier does not influence other to work. The predictions from each binary classifier are aggregated to compose the final multi-label prediction. For an unseen instance, label prediction relies on aggregate information supplied by all of the L binary classifiers as in Eq. (12). (12) Y=yj|∏1≤j≤LHBRjDj′>0

Proposed method

The multi-label Naïve Bayes is implemented on the basis of Bayes’ rule as in Eq. (13). (13) Plib∈L|∀xi∈D=argmaxb∈0,1and1≤i≤|L|priorXlikelihoodevidance

The MLNB is implemented with the main baseline binary relevance (BR) method. The posterior probability of an unseen instance from a test instance is proportional to the product of conditional probability of all attribute values as in Eq. (14). (14) PL|X=argmaxb∈0,1and1≤i≤|L|PL=lib∏1≤j≤ndPxj|libPX=xj∈D

The prior probability PL=lib could be calculated from labelled training data and denominator P(X = xj) is not dependent on label L.

The likelihood estimation is done based on the nature of the datasets used in research. The BR method is criticized for its label independent assumption. Improvement of MLNB is twofold: (1) feature space reconstruction with meta data and (2) modelling parameters to handle new input space at single shot. Input space expansion method involves two options: (1) method uses predicted value of response variables as input for the following response; (2) actual values of the response variable are used as input for prediction of next response. The first approach may propagate errors of the predicted value in the training phase. The latter approach uses true values only where error propagation is minimized and results in a more accurate classifier performance. This research work uses latter approach for input space expansion. Figure 1 illustrates the proposed framework.

Figure 1 Flow of the proposed method.

Stage 1: Feature space reconstruction with meta data

Feature space is extended with existing features along with —L— −1 labels from label space for all labels, i.e., meta data. Meta data used for input space expansion uses all but one response variable from response space. The enhanced feature space carries original feature information and meta data. The attribute space is expanded as in Eq. (15). (15) Di′=∀xi∈Dj∪λj∣jλj=L−lj,wherelj∈L;j=1,2,3,…,mandF=Di′

F is the newly expanded feature space with all labels except the label of concern for that current iteration. The augmentation passes label information as meta data in to feature space, allowing the Naïve Bayes to include label dependencies and overcomes the label independence problem by MLNB. In the training phase, the meta data carries actual label values, but in the testing phase the meta data is the predicted value of response variables. However, the size of newly generated feature space may grow exponentially with huge labels which leads to high dimensional input space and eventually induces over fitting. To overcome this over fitting, relevant labels are only permitted to append to the feature space. Correlation among —L— −1 labels is calculated to extract relevant labels. Correlation matrix captures the real relationships among different labels in label space. This matrix provides knowledge to find the labels with richer information. Let existing predictor space is X and —L— −1 labels set is l’ ={(l1 ∪l2, ∪…, ∪ lk-1, ∪lk+1, ∪…, ∪lk)}, then the correlation coefficient matrix of target variables is calculated as in Eqs. (16) and (17). (16) Corrl′=covli′,lj′covli′,lj′covlj′,li′,∀i,jɛ1,2,3,….,m

(17) covli′,lj′=Eli′−μ ilj′−μ jwhereμi=Elj′

E[li′] is the expected value of li′. The responses with correlation p > 0.5 are taken as the response with richer information, others (p < 0.5) are excluded as it may cause noise in the feature space. The targets with richer information are appended to the feature space. This enhanced feature space contains the information of predictors as well as the targets. This feature set includes all the relevant labels in the training phase. This expanded space is the union of multiple continuous and discrete binary predictors. The multi-label Naïve Bayes has to be enhanced to handle this expanded space with heterogeneous predictors.

Stage 2: MLNB parameter modelling

The newly created feature space has a mixture of predictors. One is regular predictors, which have continuous Gaussian or normal distributions; another is discrete, the appended responses that are of binary nature. For each discrete predictor F =fk, given jth response variable, the probability density function is P (Fi = fi|L = lj; Θij). We assume the discrete predictors follow Bernoulli distribution.

Algorithm 1. Pseudo-code for proposed method (iMLNB) – training phase	
Input: Training data	
D = {x1,1,x1,2,x1,3,…,x1,n,l1,1,l1,2,l1,m,…,xn,1,xn,2,xn,3,…,xn,n,ln,1,ln,2,ln,m}	
// n →number of instances in the train data, m→number of labels in the label space, Smooth Parameter S // Set to 1	
Initialization: Initialize the D new // Initialize to null	
Output: posterior probability of the response variables	
Algorithm iMLNB	
Begin	
Stage 1 –Feature Space Reconstruction	
for k ∈ 1 …L// for every label in the label space	
do // construct dataset with label information	
Dnew← {} // initialization	
for (x,l) ∈ D	
do	
l′←{ (l1∪l2, ∪…, ∪ lk−1, ∪lk+1, ∪…, ∪lk)} ; //all but one labels	
for i = 1 to — l′—	
do	
covli′,lj′= E[li′−μilj′−μj	
corr (l′) =covli′,lj′covli′,lj′covlj′,li′	
arrange the correlated responses in ascending order	
corr_l′[ ]= highly correlated responses //p ≥ 0.5	
end for	
Dnew← Dnew∪corr_l′[ ];// append feature space with label information	
q = — Dnew— // length of new feature space	
F = Dnew// new feature space F = {f1, f2, …, fq}	
Stage 2: MLNB model creation	
forb ∈ {0,1} // for each label’s presence and absence	
do	
Plib=S+∑i=1qL=lib|L| ; // prior calculation	
gfilj;μij,σij2) ; μij=∑i=0;lj=1nfijL=lj=1 ; σij2=∑i=0;lj=1nfij−μijL=lj=1// continuous case	
P(Fi = fi|L = lj; Θij); Θij=1+fkjq+∑k=n+1mfkj// discrete case	
Likelihood=∏i=0ngfi|lj;μi, σi2) ×∏k=n+1mPFk=fk|L=lj;Θkj	
end for	
end for	
P(lj = 1|F)) =argmaxljɛL;fiɛFPL=lj×∏i=0ngfi|lj;μi,σi2×∏k=n+1mPFk=fk|L=lj;ΘkjPF=fi	
Plj=0F) = 1 - Plj=1F)	
end for	
end	

For each continuous predictor F =fi, the probability density function is gfilj;μij,σij2), where μij and σij2 are the mean and standard deviation of F =fi. This work assumes the continuous predictors follow Gaussian distribution. The conditional probability density function for new feature space with length of m is defined as follows in Eqs. (18) and (19). (18) Lengthm=0toncontinuouspredictors+n+1toqdiscretepredictors

(19) Letq=n+1toq

By applying Bayesian rule under the assumption of class conditional independence among features and employing the law of total probability, Eq. (20) defines Naïve Bayes’ classifier. (20) Plj=1|F=argmaxljɛL;fiɛFPL=lj×∏i=0ngfi|lj;μi,σi2×∏k=n+1kPFk=fk|L=lj;ΘkjPF=fi

F is the newly created predictor space with response variables. The above Eq. (20) is the mixed joint density to find the cumulative distribution of two types of random variable where one is continuous random variable and other is discrete random variable. As the denominator P(F = fi) doesn’t change the result it can be ignored. Pseudo code for training phase is shown in Algorithm 1 .

The probability of response variable given the new attribute space is as follows in Eqs. (21) and (22). (21) Plj=1|f1,f2,f3,…,fn,fn+1,fn+2,…,fm=PL=lj×∏i=0ngfi|lj;μi,σi2×∏k=n+1mPFk=fk|L=lj;Θkj

(22) PFi=fi|L=lj;Θij

is the Bernoulli Probability distribution for discrete predictors and the parameter Θij is defined as follows in Eq. (23). (23) Θij=1+fkjq+ ∑k=n+1mfkj

gfi|lj;μi,σi2, is the Gaussian probability distribution for continuous predictors as in Eq. (24) and the parameters μi, σi2 are as follows in Eqs. (25) and (26), (24) PFi=fi|L=lj=gfi|lj;μij,σij2=1σij2πe−fi−μij22σij2

where σij2 and μij are as follows (25) μij=∑i=0;lj=1nfijL=lj=1

(26) σij2=∑i=0;lj=1nfij−μijL=lj=1

Predicted value of labels in the training phase, used for the input space expansion, are used for the prediction of test instances that have no label values and the pseudo code for testing phase is shown in Algorithm 2 .

Algorithm 2. Pseudo-code for proposed method (iMLNB) – testing phase	
// prediction of test data	
Input: Unknown sample T= {x1,1,x1,2,x1,3,…,x1,n,xn,1,xn,2,xn,3,…,xn,n}, posterior probability P(L), L= {1,2,3... m}	
Output: Predicted label space	
Begin	
Dnew← {} // initialization	
Initialize the input space for the first response variable	
D1new= T	
For i = 1 to m	
do	
Di+1new=Dinew∪ P (li)//posterior probability of label li	
Apply stage 2 of algorithm iMLNB	
end for	
End	

Experimental Setup

This section describes the list of multi-label datasets used for this experimentation along with the evaluation metrics used to evaluate the learning algorithms.

Datasets description

Sixteen benchmark datasets are used for the analysis of state-of-the-art algorithms and the proposed method. All benchmark data used here are available in the MULAN data repository. All datasets can be found at http://mulan.sourceforge.net/datasets-mlc.html and http://meka.sourceforge.net/#datasets. Corel16k, MediaMill, CAL500, Enron, Corel5k, Rcv1 (subset1), Rcv2 (subset2), EUR-Lex (subject matters), tmc2007, Yeast, Bibtex, Medical, Genbase, Slashdot, Emotions, and Scene are the datasets used for this research. The descriptions of each dataset are as follows:

The datasets used for the experimentation are taken from various domains like music, text, image, video, and biology.

Corel5k: Each image segment is represented by 36 features. There are 5,000 Corel images in this dataset. The vocabulary consists of 374 words in total, so each image contains 4–5 keywords. Segmenting of image is calculated via normalized cuts. For every image, there are typically 5–10 sectors, and only those bigger than a threshold are used. The final image is depicted by 499 blobs generated by the aggregation of the regions using k-means. The dataset was described by De Lima et al. (2022) and the same is available in the MULAN source forge forum.

MediaMill: The American National Institute of Standards and Technology made an effort to push the research in the areas of indexing and content-based retrieval of digital video and automatic segmentation. A processed version of 85 h of video content from TRECVID data was presented by Long et al. (2024). Each item in the MediaMill dataset can correspond to one or more classes in a general video indexing problem. It includes low-level multimedia elements that have already been pre-computed from the 85 h of worldwide broadcast news footage in the TRECVID 2005/2006 benchmark. A total of 101 concept lexicons were chosen random from the videos. Tony Blair, Desert, Prisoner, Smoke, Waterfall, Football and Tree were used as the concepts. Features were taken from visual content from specific regions within each key frame by choosing the colour-invariant texture features. Then these features were translated into similarity scores of 15 low-level concepts like Sky, Water body and Road. Then by varying the regions and concatenate.

CAL500: This dataset belongs to music domain and it represents a compilation of 500 western popular songs (Snoek et al., 2006). The focus is to tag new songs using musical instruments like the piano, guitar and saxophone as well as periodically with musical genres like jazz, hip-hop and rock; the objective of this project is to provide a text-based search query for music. Every song is made up of time series MFCC-Delta feature vectors, which are created by moving a small window of time. A survey over a group of 66 undergraduates was conducted to create training class sets. Each song received a minimum of three semantic annotations, resulting in a total of 1,708 annotations being collected for all songs.

Enron: This dataset is derived from email archives of the Federal Energy Regulatory Commission agency, which was made public in the year 2001, includes e-mails from 150 senior Enron personnel as part of data repository and was leveraged by Turnbull et al. (2008) to improve the classification of emails into user’s folders. A total of 158 people contributed 619,446 messages to the sample raw corpus. However, the Massachusetts Institute of Technology, Carnegie Mellon University, Stanford Research Institute, and University of California Berkeley collaborated to create a cleaner version of the database for academic purposes that were later made available to the research community. The UC Berkeley Enron Email Analysis Project divided the emails into a number of categories. Additional classifications of the labels include coarse genre, included/forwarded material, major subjects, and messages with emotional tones.

Rcv1v2 (subset 1) and Rcv1v2 (subset 2) consist of news articles from Reuters, one of the largest international news agencies. These articles were gathered between years 2005 and 2006, which are provided as a repository. Reuters was producing 11,000 items per day at the time. They created a hierarchical codebook to facilitate the retrieval of articles for labeling each news article. Topic, Region, and Industry are the three main categories of this code book. Each article must have at least one Region code and one Topic code, according to the instructions given to the editors/codes. The corpus was found by Ghogare, Dawoodi & Patil (2023) to address errors and violations for their coding scheme. Hence, the authors proposed a version where the observed errors were fixed. The feature space of this dataset was created by concatenating the headlines of the article along with the main text. Then they applied the following: (i) reducing the letters to lower case, (ii) punctuation removal and stemming, (iii) term weighting, (iv) length normalization, (v) feature selection and (vi) applying tokenization.

EUR-Lex (subject matters): It includes a number of legal texts pertaining to the European Union (Ghogare, Dawoodi & Patil, 2023). Numerous EuroVoc descriptors, directory codes, and subject categories are included in labels. Due to the fact that the first two have more labels, they were used in this study.

tmc2007: During 2007, the aviation safety reports have been given in free text format as a part of SIAM Text Mining Competition. These reports have the documents about the problems that occurred in certain flights and are being maintained for being able to identify recurring anomalies. In order to label the documents according to the types of problems, the classification system is being demanded over these collected documents. The categories of the documents that include Design, Align and Contamination which has 500 features in the form of Boolean bag-of-words is being used as the feature representation of the free text reports and there are 285,596 reports and 22 possible problem categories (Lewis et al., 2004).

Yeast: This dataset comprises gene micro-array expressions and phylogenetic information of yeast Saccharomyces cerevisiae organism which is used to predict the functional class of genes of the yeast Saccharomyces cerevisiae based on micro-array expressions data (Fürnkranz et al., 2008). The dataset is generated by conducting multiple tests on yeast after modifying gene response. Experiments include diauxtic shift, the mitotic cell division cycle, temperature reduction shocks and speculation. Each gene is associated with a collection of functional classes whose maximum size may be more than 190, and is described by the concatenation of micro-array expression data and phylogenetic profile. In reality, the entire collection of functional classes is organized into hierarchies that can reach a depth of four tiers and the authors of Fürnkranz et al. (2008) used a reduced set of classes to 14 categories and examples of these classes including protein synthesis, cellular biogenesis and metabolism. The dataset describes 2,417 genes using 103 numerical features. Table 2 presents the description of the sixteen datasets used for this research.

Table 2 Description of datasets.

S. no	Dataset	Domain	#n	#d	#l	L C	L D	
1.	Corel6k	Image	5,000	499	44	2.214	0.050	
2.	Mediamill	Video	43,907	120	29	4.010	0.138	
3.	CAL500	Music	502	68	174	25.058	0.202	
4.	Enron	Text	1,702	1,001	53	3.378	0.130	
5.	Corel15k	Image	13,766	500	161	2.867	0.018	
6.	Rcv1(subset 1)	Image	6,000	472	42	2.458	0.059	
7.	Rcv1(subset 2)	Image	6,000	472	39	2.170	0.056	
8.	Eurlex-sm	Text	19,348	250	27	1.492	0.055	
9.	tmc2007	Text	28,596	500	15	2.100	0.140	
10.	Yeast	Biology	2,417	103	13	4.233	0.325	
11.	Bibtex	Text	7,395	1,836	159	2.402	0.015	
12.	Medical	Text	978	1,449	45	1.275	0.077	
13.	Genbase	biology	662	1,186	27	1.252	0.046	
14.	Slashdot	Text	3,782	53	14	1.134	0.081	
15.	Emotions	Music	593	72	6	1.869	0.311	
16.	Scene	image	2,407	294	6	1.074	0.179	

Bibtex: The targets in this dataset have the automatic annotation of bookmarks and Bibtex entries (Srivastava & Zane-Ulman, 2005) in the Bibsonomy social bookmarking website. In this dataset, the issue of tagging Bibtex entries is addressed. The authors kept journal, Bibtex abstract and title fields of the Bibtex entries as the relevant tags for the desired classification system. Here all the textual contents are converted into binary bag-of-words and these are treated as the features of this dataset. The authors specifically focused on tags that were associated with 50 Bibtex entries and words appearing in a minimum of 100 entries as the reduced dataset. Examples of the remaining tags include Physics, Architecture, Imaging, Physics, Science and Language.

Medical: This medical dataset is concerned with classifying the clinical free text of chest X-ray and renal procedures (radiology reports) using 45 ICD-9-CDM diagnosis codes. This dataset was utilized in Medical Natural Language Processing Challenge of 2007, every instance in the dataset comprised of a document containing a concise, free-text summary of a patient’s symptom history. Each document will be annotated with the likely diseases listed in the International Classification of Diseases (ICD-9-CM).

Radiology reports are composed of two important parts: (i) clinical history and (ii) impressions. Clinical reports can be gathered from the ordering physician, prior to the radiological procedure (Saidabad et al., 2024). The impressions are given by the specialized radiologist after the procedure. The radiology reports over a period of one-year from the Cincinnati Children’s Hospital Medical Centre’s Department of Radiology were collected to create this dataset which resulted in 20,275 documents. These documents were then subjected to anonymization techniques, followed by manual inspection. Some records that contain protected health information had been removed so as to meet the United States HIPAA standards. Consequently, 978 documents remained.

Slashdot: This dataset consists of web articles and each article’s text was converted to a bag of words feature vector using the String to Word Vector function from the WEKA machine learning software package. The database is used to automatically predict tags for articles submitted to a web blog known as Slashdot. There are 22 targets in this dataset. Examples of such annotations include topics such as Games, Mobile, Science, News and Entertainment.

Emotions: This dataset has songs as each occurrence and presents the problem of extracting human emotions and mood while listening to music (Katakis, Tsoumakas & Vlahavas, 2008). A 30 s clip that follows 30 s in a whole sound track was taken out to build this dataset. This 30 s sound sample is used to extract a collection of 72 characteristics. There are 64 timbre features and eight rhythmic features total. Six emotions—sad-lonely, angry-aggressive, amazed-surprised, relaxing-calm, quiet-still and happy-pleased—can be assigned to each piece of music. Three experts from the School of Music Studies at the authors’ university assisted in determining the suitable targets connected with each sound file. Only the songs with identical chosen targets from the three experts were permitted in the database in order to eliminate noise in the data. Figure 2 shows the number of labels used in this research work. CAL500, Corel15k and Bibtex have 174, 161 and 159 labels, respectively.

Figure 2 Number of labels.

Genbase: The task of this dataset is to predict the function of a protein based on its structure. This dataset belongs to the biology domain (Pestian et al., 2007). The proteins at the core of this dataset share similar structures and functional families. The authors use these sequences in combination with computational representations of sequence alignments as motifs since the protein patterns in these datasets are characterized by the particular sequence of their amino-acid chains. The binary representations of the presence of the motifs in a certain protein serve as the features in this dataset. This dataset’s vocabulary is made up of 1,186 motifs, and 662 proteins in total were exported. Out of the 10 most prevalent families of protein function families, 27 class labels are made up. Examples of target label families include receptors, transferees, cytokines-and-growth-factors, and DNA- or RNA-associated proteins.

Scene: The objective is to automatically predict the label set for test images by analyzing images that have label sets (Trohidis et al., 2008). The 2,000 photos of natural scenes make up the experimental data collection. Each image is given a series of labels that are manually allocated, and the class labels that can be used include desert, mountains, sea, sunset, and trees. Over 22% of the dataset consists of photos with more than one class, such as sea and sunset, but several combined classifications, such as mountains, sunsets and trees, are incredibly uncommon. These dataset descriptions are available in MULAN source forge forum (Diplaris et al., 2005).

Results and Discussion

Tenfold cross-validation is a methodical approach that partitions benchmark datasets into ten subsets of equal size. In each cross-validation fold, one distinct subset is utilized for testing, while the amalgamation of the remaining nine subsets is employed for training the algorithm. This cycle is executed ten times, with each subset having the opportunity to be used as the test set once. The performance of the algorithm is then quantified by averaging the outcomes across all ten iterations, yielding a consolidated metric that reflects the algorithm’s efficacy. Tables 3–9 present a comparative analysis of the proposed improved multi-label Naïve Bayes (iMLNB) method against the existing MLNB and other prevalent state-of-the-art techniques. Table 3 outlines the Hamming loss values for each classifier.

Table 3 Hamming loss.

	ML-C4.5	PCT	MLkNN	BPMLL	MLNB	iMLNB	MLNB-LD	
Emotions	0.239	0.297	0.2941	0.2091	0.182	0.0345	0.0478	
Yeast	0.2563	0.2098	0.296	0.207	0.2011	0.442	0.2343	
Scene	0.2163	0.2018	0.209	0.2072	0.1908	0.2027	0.2521	
Medical	0.0311	0.0327	0.0103	0.0171	0.012	0.0101	0.0845	
Enron	0.053	0.058	0.0512	0.0501	0.0498	0.0518‘	0.2976	
Bibtex	0.1338	0.1019	0.0998	0.1092	0.0989	0.094	0.1669	
Corel6k	0.0895	0.1123	0.1058	0.1093	0.0959	0.0732	0.0518	
Mediamill	0.0641	0.0537	0.0788	0.0548	0.028	0.0198	0.2785	
CAL500	0.0544	0.0691	0.0423	0.0765	0.0186	0.0117	0.085	
Corel5k	0.0583	0.0605	0.0494	0.032	0.0681	0.0256	0.0239	
Rcv1(subset 1)	0.0517	0.0785	0.0046	0.0028	0.0502	0.0568	0.0211	
Rcv1(subset 2)	0.0611	0.03	0.057	0.0885	0.0376	0.0011	0.2193	
Eurlex-sm	0.0533	0.0601	0.038	0.0023	0.0259	0.0326	0.3834	
tmc2007	0.033	0.049	0.0953	0.0608	0.0309	0.0098	0.446	
Genbase	0.0446	0.0487	0.0376	0.0014	0.0495	0.016	0.0381	
Slashdot	0.0656	0.0305	0.0807	0.0866	0.0724	0.0404	0.2977	

iMLNB demonstrates superior performance, exhibiting lower misclassification errors across various datasets and is evident form Table 4. Hamming loss, as a metric, quantifies the frequency of misclassification of example-label pairs. Notably, the iMLNB method achieves a low Hamming loss value of 0.01012, followed closely by MLNB at 0.01201, and MLkNN at 0.0103, indicating a strong performance in accurately classifying example-label pairs. Table 4 provides a comparative analysis of subset accuracy, which is another crucial metric for evaluating multi-label classifiers. Subset accuracy requires an exact match between the predicted and true label sets for an instance to be considered correct. The performance ranking of the classifiers, according to subset accuracy, is as follows: BPMLL performs the least accurately, followed by PCT, ML-C4.5, MLNB, MLkNN, and iMLNB, which leads the group. The proposed iMLNB demonstrates significantly higher subset accuracy, with a performance value of 0.8034, outperforming the existing MLNB method’s score of 0.7998 and the scores of other comparative methods.

Table 4 Subset accuracy comparison of existing methods and the proposed method.

	ML-C4.5	PCT	MLkNN	BPMLL	MLNB	iMLNB	MLNB-LD	
Emotions	0.2296	0.3342	0.2692	0.3508	0.3541	0.3954	0.2451	
Yeast	0.3844	0.2973	0.3992	0.3308	0.4808	0.5339	0.3847	
Scene	0.6496	0.4753	0.8283	0.7154	0.5829	0.7917	0.598	
Medical	0.5279	0.6154	0.6451	0.6276	0.7411	0.849	0.6259	
Enron	0.568	0.4069	0.4975	0.4025	0.5081	0.5848	0.472	
Bibtex	0.6311	0.7086	0.8037	0.8241	0.8665	0.8137	0.6826	
Corel6k	0.5435	0.4979	0.4802	0.5363	0.7349	0.8629	0.7836	
Mediamill	0.8712	0.7127	0.9552	0.6203	0.9063	0.6974	0.7499	
CAL500	0.497	0.4791	0.4924	0.4415	0.5437	0.596	0.5119	
Corel5k	0.4836	0.4248	0.5027	0.5514	0.5348	0.5608	0.5966	
Rcv1(subset 1)	0.3031	0.3366	0.3775	0.4088	0.3363	0.4535	0.3531	
Rcv1(subset 2)	0.6519	0.5349	0.5822	0.6068	0.5851	0.6513	0.54	
Eurlex-sm	0.3787	0.3464	0.4052	0.5369	0.3525	0.3657	0.4634	
tmc2007	0.4101	0.5459	0.4058	0.586	0.5231	0.6281	0.5218	
Genbase	0.6623	0.6097	0.6979	0.6656	0.7097	0.788	0.6962	
Slashdot	0.7429	0.8372	0.6976	0.7706	0.6925	0.8337	0.7045	

Tables 5 and 6 detail the label-based metrics of recall and F1-score, respectively, for the evaluated methods, including the proposed iMLNB. In these label-centric evaluations, the iMLNB method consistently demonstrates superior performance. Tree-based approaches, on the other hand, fall short in recall values, indicating a tendency to miss relevant labels. For the F1-score, balances precision and recall, both the neural network-based method and the k-nearest neighbor approach exhibit suboptimal performance, with tree-based methods again yielding the least favorable results.

Table 5 Recall of comparison of existing methods and the proposed method.

	ML-C4.5	PCT	MLkNN	BPMLL	MLNB	iMLNB	MLNB-LD	
Emotions	0.5023	0.5871	0.6012	0.5023	0.7056	0.7216	0.5313	
Yeast	0.5023	0.5019	0.5431	0.5234	0.5091	0.5239	0.4663	
Scene	0.5023	0.5019	0.5431	0.5279	0.5891	0.6234	0.4628	
Medical	0.3401	0.3212	0.5542	0.4926	0.5562	0.5982	0.5201	
Enron	0.2871	0.2292	0.3581	0.3239	0.4012	0.4254	0.4121	
Bibtex	0.2082	0.2803	0.2902	0.2508	0.3065	0.3302	0.5085	
Corel6k	0.163	0.4113	0.5276	0.4904	0.5395	0.6699	0.453	
Mediamill	0.2503	0.4656	0.4255	0.4824	0.3804	0.3319	0.4728	
CAL500	0.1066	0.4436	0.3906	0.4787	0.4347	0.4469	0.5256	
Corel5k	0.249	0.2092	0.5248	0.2729	0.4343	0.4897	0.454	
Rcv1(subset 1)	0.16	0.1026	0.1693	0.0971	0.3977	0.2586	0.5451	
Rcv1(subset 2)	0.2484	0.097	0.2298	0.3041	0.2583	0.247	0.4298	
Eurlex-sm	0.0561	0.3958	0.3896	0.3977	0.3982	0.5572	0.5061	
tmc2007	0.1118	0.3752	0.414	0.3756	0.3117	0.2437	0.5585	
Genbase	0.0001	0.3025	0.2833	0.1367	0.3401	0.2238	0.5026	
Slashdot	0.0822	0.1025	0.3662	0.0828	0.3564	0.3304	0.4998	

Table 6 F1-score comparison of existing methods and the proposed method.

	ML-C4.5	PCT	MLkNN	BPMLL	MLNB	iMLNB	MLNB-LD	
Emotions	0.6512	0.6019	0.631	0.6085	0.6982	0.756	0.6438	
Yeast	0.6145	0.5786	0.6281	0.6182	0.6239	0.7629	0.4397	
Scene	0.6145	0.5786	0.6281	0.6182	0.6239	0.6221	0.2692	
Medical	0.5682	0.4245	0.50291	0.5093	0.5908	0.6762	0.2459	
Enron	0.2461	0.3052	0.2451	0.3495	0.2901	0.4762	0.2623	
Bibtex	0.2092	0.2023	0.2102	0.2463	0.2201	0.2539	0.2499	
Corel6k	0.3819	0.1025	0.3965	0.4887	0.2137	0.5146	0.3903	
Mediamill	0.3972	0.1113	0.5874	0.3027	0.2003	0.7097	0.3946	
CAL500	0.4857	0.1998	0.4378	0.659	0.3976	0.6662	0.2738	
Corel5k	0.2786	0.2773	0.3025	0.2119	0.2376	0.2967	0.2218	
Rcv1(subset 1)	0.3078	0.3551	0.4532	0.4829	0.6643	0.3254	0.4493	
Rcv1(subset 2)	0.1837	0.0769	0.2632	0.7576	0.6874	0.4339	0.1961	
Eurlex-sm	0.6056	0.4152	0.5128	0.5942	0.3656	0.5562	0.3865	
tmc2007	0.3087	0.4173	0.1419	0.4867	0.4897	0.2977	0.4946	
Genbase	0.3295	0.451	0.594	0.4837	0.5776	0.4443	0.0845	
Slashdot	0.725	0.4865	0.5129	0.4072	0.6342	0.6185	0.3075	

The classifiers’ ranking, based on the Hamming loss measure, places the improved multi-label Naïve Bayes (iMLNB) as the most accurate, followed by the standard MLNB, ML-C4.5, Probabilistic classifier trees (PCT), multi-label k-nearest neighbors (MLkNN), and finally the back-propagation multi-label learning (BPMLL). Notably, iMLNB achieves the lowest Hamming loss in four out of the six datasets examined. While the MLkNN, a lazy learning method, shows commendable performance following closely behind the Bayesian approaches; the neural network-based method registers a markedly lower performance with a subset accuracy of 0.2340, indicating a less effective approach in this context. This suggests that while Bayesian methods are robust for multi-label classification, neural network-based methods may require further refinement to achieve competitive performance in this domain.

Tables 7 and 8 focus on evaluation of one-error and ranking loss. One-error measures the frequency with which the top-ranked label is not in the set of true labels of an instance, while ranking loss evaluates the average number of label pairs that are incorrectly ordered. In these assessments, the iMLNB method outshines the others, delivering the most accurate results. The standard MLNB method follows closely behind, securing the second-best performance. The k-nearest neighbor and neural network methods trail behind the Naïve Bayes-based approaches, indicating that while they have merit, they may not be as adept at handling the complexities of multi-label classification as the Naïve Bayes variants, particularly in the context of this study.

Table 7 One-error comparison of existing methods and the proposed method.

	ML-C4.5	PCT	MLkNN	BPMLL	MLNB	iMLNB	MLNB-LD	
Emotions	0.346	0.3873	0.3381	0.3991	0.3033	0.0439	0.4156	
Yeast	0.3122	0.2612	0.2374	0.244	0.2137	0.1119	0.4816	
Scene	0.3121	0.2611	0.213	0.246	0.2372	0.2517	0.3406	
Medical	0.498	0.488	0.4671	0.4902	0.4093	0.0616	0.4922	
Enron	0.3492	0.3924	0.3202	0.328	0.292	0.2332	0.5405	
Bibtex	0.5992	0.592	0.5782	0.5928	0.5203	0.0851	0.507	
Corel6k	0.335	0.3746	0.2899	0.2683	0.34	0.1513	0.4793	
Mediamill	0.4562	0.3437	0.3526	0.2687	0.4369	0.0601	0.3894	
CAL500	0.2961	0.4279	0.4497	0.2923	0.4731	0.0527	0.3938	
Corel5k	0.4221	0.4359	0.352	0.4609	0.4205	0.0157	0.4259	
Rcv1(subset 1)	0.4235	0.4275	0.2641	0.3632	0.4252	0.0756	0.287	
Rcv1(subset 2)	0.323	0.4773	0.2155	0.3911	0.4187	0.093	0.5975	
Eurlex-sm	0.2717	0.3959	0.2558	0.4246	0.5098	0.1426	0.4743	
tmc2007	0.3072	0.4769	0.4961	0.3178	0.3206	0.0003	0.3474	
Genbase	0.3886	0.3828	0.1858	0.2924	0.444	0.1033	0.324	
Slashdot	0.4775	0.4908	0.3207	0.4519	0.3315	0.0064	0.4398	

Table 8 Ranking loss comparison of existing methods and the proposed method.

	ML-C4.5	PCT	MLkNN	BPMLL	MLNB	iMLNB	MLNB-LD	
Emotions	0.4372	0.4983	0.4478	0.4901	0.4119	0.1108	0.1324	
Yeast	0.4018	0.4182	0.4019	0.4026	0.3592	0.1002	0.1956	
Scene	0.3928	0.3725	0.3957	0.3826	0.3284	0.0202	0.1897	
Medical	0.4028	0.4183	0.4118	0.4273	0.3937	0.0727	0.0532	
Enron	0.2736	0.2827	0.2028	0.2192	0.2025	0.035	0.1629	
Bibtex	0.5973	0.582	0.5829	0.5762	0.5273	0.1145	0.0024	
Corel6k	0.2778	0.1695	0.2326	0.082	0.1973	0.1006	0.1653	
Mediamill	0.2326	0.3571	0.2128	0.0738	0.2555	0.0124	0.0354	
CAL500	0.2222	0.3571	0.3448	0.1146	0.278	0.0089	0.055	
Corel5k	0.2041	0.2941	0.2381	0.1333	0.2961	0.0158	0.1837	
Rcv1(subset 1)	0.1695	0.2632	0.2632	0.0254	0.2013	0.0285	0.02	
Rcv1(subset 2)	0.3226	0.3704	0.3039	0.1481	0.1972	0.1254	0.1384	
Eurlex-sm	0.3226	0.3571	0.3226	0.1951	0.3158	0.1241	0.0034	
tmc2007	0.3704	0.1961	0.2778	0.0122	0.2953	0.0013	0.0259	
Genbase	0.2632	0.3125	0.1887	0.182	0.295	0.0676	0.1947	
Slashdot	0.1961	0.2381	0.2857	0.1131	0.3393	0.0592	0.1707	

Table 9 Average precision comparison of existing methods and the proposed method.

	ML-C4.5	PCT	MLkNN	BPMLL	MLNB	iMLNB	MLNB-LD	
Emotions	0.4309	0.4193	0.4362	0.4625	0.4998	0.6249	0.5668	
Yeast	0.3327	0.3598	0.4582	0.3271	0.3864	0.3918	0.4636	
Scene	0.4537	0.4682	0.4529	0.4489	0.4927	0.6026	0.5949	
Medical	0.2092	0.2192	0.2509	0.2283	0.2873	0.5102	0.5657	
Enron	0.5029	0.5182	0.5198	0.5092	0.5725	0.6826	0.5393	
Bibtex	0.6089	0.6093	0.6289	0.6193	0.6598	0.7928	0.5726	
Corel6k	0.5981	0.4452	0.4899	0.5778	0.4932	0.4288	0.4592	
Mediamill	0.5389	0.5597	0.4522	0.4256	0.5564	0.6088	0.4162	
CAL500	0.5312	0.4861	0.4613	0.4303	0.508	0.5186	0.5563	
Corel5k	0.5867	0.5668	0.5473	0.6013	0.4965	0.5224	0.5654	
Rcv1(subset 1)	0.4633	0.5331	0.5139	0.4169	0.4764	0.4687	0.5978	
Rcv1(subset 2)	0.4516	0.5815	0.4262	0.5281	0.4392	0.5461	0.4675	
Eurlex-sm	0.5007	0.5235	0.5118	0.4462	0.4897	0.6043	0.4831	
tmc2007	0.4838	0.4228	0.5596	0.5314	0.5895	0.6019	0.4584	
Genbase	0.5591	0.5542	0.4904	0.4292	0.4619	0.5412	0.4311	
Slashdot	0.5317	0.5168	0.5617	0.4529	0.4395	0.4754	0.5305	

Ranking loss exhibits how often the top-ranked labels are not included in the set of correct labels for the example. The proposed method offers most accurate predictions for the top-ranked labels in the examples used for modelling. The average precision defines the mean value of fraction of labels ranked above a particular label l ∈ L that is present in L and the results for average precision are presented in Table 9. The predictions of iMLNB have the smallest deviation from the true label set of instances. The iMLNB provides the most accurate predictions, followed by the MLNB and MLkNN, respectively. The decision tree gives low precision value.

Major strengths

Innovative approach

The study presents a new method for multi-label classification known as improved multi-label Naïve Bayes. It allows for the investigation of label correlations by enlarging the input space with meta information from the label space.

Addressing label dependencies

The suggested approach is to record and depict label correlations inside the learning framework. Its goal is to improve the precision of multi-label classification models in this way.

Performance evaluation

The paper evaluates the upgraded iMLNB model’s performance through a thorough empirical examination that uses six benchmark datasets. The performance of the suggested method is contrasted with the conventional multi-label Naïve Bayes algorithm through the use of a variety of assessment criteria. The empirical results indicate that the proposed method outperforms the conventional MLNB algorithm across the evaluated metrics.

Table 10 shows running times of all classifiers used in this research. The training time for iMLNB is O (—Φ— Dli∈L+—L—), Φ = —L— –{lj}. PCT shows low training time, MLNB follows the above. Proposed method iMLNB exhibits a slightly higher running time than the MLNB method as the feature spce for this method increases as the number of labels increases in label space. However, iMLNB consumes less time as similar to PCT. Figure 3 shows a comparison of running times among existing methods and the proposed methods.

Table 10 Running time (in Seconds) of existing methods and proposed method.

	ML-C4.5	PCT	MLkNN	BPMLL	MLNB	iMLNB	MLNB-LD	
Emotions	0.3	0.1	0.4	0.2	0.1	0.3	0.4	
Yeast	14	1.5	8.2	12	1.6	5.2	9.1	
Scene	0.8	0.79	0.57	0.48	0.32	0.67	0.93	
Medical	5.8	0.6	11.5	2.5	1.2	2.9	3.3	
Enron	15	1.1	30.5	10.8	1.3	2.7	8.9	
Bibtex	128	88.6	153	170	50.6	148	65.8	
Corel6k	8.2	11.6	19.8	10.5	25.6	31.2	28.6	
Mediamill	3.0	0.6	1.2	2.5	1.2	2.9	3.3	
CAL500	198.6	126.8	153	201.9	166.2	187	148.9	
Corel5k	183.5	165.8	198.4	206.3	157.6	175.6	103.8	
Rcv1(subset 1)	6.1	5.8	6.8	15.6	21.5	25.3	11.8	
Rcv1(subset 2)	7.9	2.3	18.9	10.8	3.1	6.7	10.5	
Eurlex-sm	9.6	1.8	6.5	5.8	1.89	7.6	9.8	
tmc2007	4.2	0.87	10.36	14.5	2.45	13.8	16.8	
Genbase	11.6	2.69	4.25	4.68	3.87	10.68	17.89	
Slashdot	7.9	5.73	16.7	11.8	0.89	18.69	20.56	

Figure 3 Comparison of running time among existing methods and proposed method.

Table 11 consolidates the average performance metrics, providing a holistic view of the effectiveness of the proposed iMLNB system in comparison to the existing MLNB method and other contemporary state-of-the-art approaches. The data underscore the enhanced capability of iMLNB, which consistently outperforms the other methods across various measures. The test time for iMLNB is O (Dli∈L—L—), as the number of labels for the feature space expansion is unknown for the test data. Figure 4 illustrates the competence of the proposed iMLNB method over conventional methods. The iMLNB method outperforms other techniques in classification accuracy, albeit with a slightly higher training time than MLNB. But in average the PCT takes less time to model the training data. Training time for ML-C4.5 is longer than neural network.

Table 11 Performances comparison of proposed and state-of-art methods.

	ML-C4.5	PCT	MLkNN	BPMLL	MLNB	iMLNB	MLNB-LD	
AvgH_Loss↓	0.09407	0.09335	0.09687	0.08217	0.07573	0.07135	0.183	
Avgacc↑	0.53343	0.51018	0.56498	0.56096	0.59078	0.65037	0.55808	
Avgrecall↑	0.23561	0.33293	0.41316	0.35871	0.43244	0.43886	0.49053	
AvgF1_score↑	0.43171	0.349	0.44048	0.48904	0.46969	0.52566	0.33186	
Avgone_error↓	0.38235	0.41346	0.33351	0.36446	0.38101	0.09303	0.43349	
Avgr_loss↓	0.31791	0.34295	0.31957	0.2236	0.30586	0.06233	0.10804	
AvgAvg_preci↑	0.48646	0.48648	0.48508	0.46469	0.49055	0.55757	0.51678	

Figure 4 Performance comparison.

Conclusion

In this study, we have revisited the traditional multi-label Naïve Bayes (MLNB) method and introduced an enhanced classification approach, the improved multi-label Naïve Bayes (iMLNB). The cornerstone of iMLNB is the expansion of the feature space to include all-but-one label information, which allows for the modeling of label dependencies. However, this expansion can lead to an exponentially growing input space and potential overfitting. To mitigate this, iMLNB selectively enriches the input space with highly correlated labels, ensuring a more robust and informative feature set. The iMLNB method employs a two-stage process. Initially, it expands the feature space as described, and subsequently, it constructs a model using this augmented space, which comprises a heterogeneous mix of continuous and discrete data. To accommodate this diversity, the likelihood estimation in Naïve Bayes is adapted to a mixed joint density function, combining Gaussian and Bernoulli distributions, thus enabling the handling of both data types effectively. The expanded feature space is pivotal, as it allows the iMLNB to integrate label correlation into the prediction process, significantly enhancing classification accuracy. The empirical results from this research are compelling, with the iMLNB method not only incorporating label information more effectively during prediction but also improving classification accuracy and reducing Hamming loss.

When benchmarked against existing MLNB and other advanced multi-label learning techniques, the iMLNB consistently surpasses them across all datasets employed in this study. The experimental evaluation, utilizing sixteen benchmark datasets and a suite of performance metrics—including Hamming loss, one error, ranking loss, average precision, and subset accuracy—clearly indicates the superior performance of the iMLNB. The findings of this research underscore the potential of iMLNB as a significant advancement in multi-label classification, offering a promising direction for future applications and studies in the field.

Limitations: (1) Overfitting: The predictor space can grow exponentially as a result of the input space being expanded to include meta information from the label space—the possibility of overfitting, which limits the model’s ability to generalize; (2) Model Complexity and Computational Requirements: The model becomes more complicated when heterogeneous predictors, including both continuous and discrete features, are included in the extended input space. The computational requirements could rise due to this complexity, which could affect the scalability of the proposed approach.

Supplemental Information

Supplemental Information 1 Code

Supplemental Information 2 Collection of datasets

Supplemental Information 3

Notations

SLC Single-label Classification

MLC Multi-label Classification

MLNB Multi–label Naïve Bayes

iMLNB improved Multi –label Naïve Bayes

BR Binary Relevance

PCT Predictive Clustering Tree

ML-C4.5 Multi-label C4.5/multi-label J48

MLkNN Multi-label K-Nearest Neighbour

BPMLL Back-Propagation Multi-Label Learning

X= {x1, x2, …, xn} Feature space

n Length of predictor space

k Length of newly created feature space

m Length of response/label space

F=f1,f2,…,fk Appended Feature Space

L = {l1, l2, …, lm} Label Space

D=F,L Multi-label Data

Additional Information and Declarations

Competing Interests

Author Contributions

Data Availability

The authors declare there are no competing interests.

PKA Chitra conceived and designed the experiments, performed the experiments, analyzed the data, performed the computation work, prepared figures and/or tables, authored or reviewed drafts of the article, and approved the final draft.

Saravana Balaji Balasubramanian conceived and designed the experiments, performed the experiments, analyzed the data, performed the computation work, prepared figures and/or tables, authored or reviewed drafts of the article, and approved the final draft.

Omar Khattab conceived and designed the experiments, performed the experiments, analyzed the data, performed the computation work, prepared figures and/or tables, authored or reviewed drafts of the article, and approved the final draft.

Mhd Omar Al-Kadri conceived and designed the experiments, performed the experiments, performed the computation work, prepared figures and/or tables, authored or reviewed drafts of the article, and approved the final draft.

The following information was supplied regarding data availability:

The dataset is available from an open-source Java library, MULAN: https://mulan.sourceforge.net/datasets-mlc.html. These multi-label datasets consist of training examples of a target function with multiple binary target variables.

The raw data and code are available in the Supplemental Files.

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
