# Peer review of "Label dependency modeling in Multi-Label Naïve Bayes through input space expansion"

_PeerJ Computer Science, doi:10.7717/peerj-cs.2093_

## Round 0.1 · original submission · Major Revisions

Based on the reviewers' comments, the manuscript must be improved before acceptance.

Reviewer 2 has suggested that you cite specific references. You are welcome to add it/them if you believe they are relevant. However, you are not required to include these citations, and if you do not include them, this will not influence my decision.

**Language Note:** PeerJ staff have identified that the English language needs to be improved. When you prepare your next revision, please either (i) have a colleague who is proficient in English and familiar with the subject matter review your manuscript, or (ii) contact a professional editing service to review your manuscript. PeerJ can provide language editing services - you can contact us at [email protected] for pricing (be sure to provide your manuscript number and title). – PeerJ Staff

Reviewer 1 ·

Basic reporting

The paper titled Label dependency modeling in Multi-Label Naïve Bayes through input space expansion", is a novel writing. The innovation of proposed work lies in its strategic expansion of the input space, which assimilates meta-information derived from the label space, thereby engendering a composite input domain that encompasses both continuous and categorical variables is very interesting. The paper is clearly written in a good style and includes figures and tables wherever necessary.

Experimental design

The authors have clearly acknowledged and identified the contributions of their research against previous researchers' work and presents the proposed approach using rigorous empirical evaluation, utilizing six benchmark datasets.
The purpose of the paper has been very well stated in the abstract but needs clarification on the following:
1. What is the motivation for choosing iMLNB for this particular task?
2. In the discussion section, the research's strengths, limitations, and generality need more appropriate discussions.

Validity of the findings

The authors adequately evaluated their work, and all claims are clearly articulated and supported by empirical experiments

Additional comments

However, addressing the above comments would improve the quality of the paper. The overall work is good, novel and timely.

Reviewer 2 ·

Basic reporting

The work is interesting but lacks some details and clarifications

I strongly suggest improving the summary and defining just 5 keywords, there are so many keywords included that it looks like a summary of the article.

What does the "B. Notations" table contribute to the introduction and the work as a whole? The relationship and motives are not clear...

The introduction is also not clear what is new in the work (2 recent articles, 1 in 2022 and the other in 2023) out of a total of 41.

This is reflected in the following section II. LITERATURE SURVEY

In this section, I suggest including a comparative table of criteria for this work about the literature research.

I strongly suggest including at least 30% of current references (2022, 2023 and 2024).

Below are recent works with the same theme that can help authors.

https://doi.org/10.3390/bdcc6010008
doi: 10.1109/ACCESS.2023.3255164
https://content.iospress.com/articles/journal-of-intelligent-and-fuzzy-systems/ifs223218

Experimental design

I strongly suggest improving the summary and defining just 5 keywords, there are so many keywords included that it looks like a summary of the article.

Validity of the findings

I'm not sure, more concrete evidence is needed for everything previously described.

What problems do authors encounter in their research?

After all the work done, implementation, testing and results, what problems were resolved?

These contributions are not evident.

Additional comments

The work, if improved, could contribute to the magazine and readers, but as it stands, it needs to be improved.
I hope I have contributed to the authors for future improvements

·

Basic reporting

The following are some serious concerns about this paper. These may be incorporated for further correspondence,
The abstract fails to provide the necessary information about the domain. Only the expert in too specific field may grasp the actual idea. Otherwise, the general CS readers will not get the actual idea.
What is MLB? It is not defined anywhere.
Why do authors provide such in-depth details in the introduction? Again this will cause issues like abstract.
Why equations are provided in the introduction? I think the introduction should provide an overview, of needs, and contributions with any mathematical support. The necessary discussion may be added in the rest of the sections.
There is no need to highlight contributions with specific keywords.
The authors used too many abbreviations in the manuscript, hence, having difficulty understanding texts.
The methodology is well written. However, I suggest authors go through the whole network to symbols and notations.
I suggest strategies to avoid using too many of those equations without needs.
It is better to include some graphs to show the figure-wise result. It will attract readers with more attention.

Experimental design

Included in basic reportings

Validity of the findings

Included in basic reportings

Additional comments

Included in basic reportings

---

## Round 0.2 · accepted · Accept

The authors addressed all comments accurately, and the manuscript can be accepted.

Reviewer 2 ·

Basic reporting

The authors updated the work.
For final version, review text, formatting and general organization.

Experimental design

It is understood and there is a link to check the information in the repository.

Validity of the findings

It is understood and there is a link to check the information in the repository.

Additional comments

I believe that the work contributes to the magazine and readers.

·

Basic reporting

The authors have addressed all previous concerns, and hence recommended for publication.

Experimental design

Nil

Validity of the findings

Nil